# Unveiling the Function of the Mitochondrial Filament-Forming Protein LACTB in Lipid Metabolism and Cancer

**DOI:** 10.3390/cells11101703

**Published:** 2022-05-20

**Authors:** Annunziata Cascone, Maciej Lalowski, Dan Lindholm, Ove Eriksson

**Affiliations:** 1Department of Biochemistry and Developmental Biology, Faculty of Medicine, University of Helsinki, FIN-00014 Helsinki, Finland; sonia.cascone@helsinki.fi (A.C.); dan.lindholm@helsinki.fi (D.L.); 2HiLIFE, Meilahti Clinical Proteomics Core Facility, Faculty of Medicine, University of Helsinki, FIN-00014 Helsinki, Finland; maciej.lalowski@helsinki.fi; 3Minerva Foundation Institute for Medical Research, Biomedicum Helsinki 2, Tukholmankatu 8, FIN-00290 Helsinki, Finland

**Keywords:** mitochondria, LACTB, serine protease, lipid metabolism, cancer

## Abstract

LACTB is a relatively unknown mitochondrial protein structurally related to the bacterial penicillin-binding and beta-lactamase superfamily of serine proteases. LACTB has recently gained an increased interest due to its potential role in lipid metabolism and tumorigenesis. To date, around ninety studies pertaining to LACTB have been published, but the exact biochemical and cell biological function of LACTB still remain elusive. In this review, we summarise the current knowledge about LACTB with particular attention to the implications of the recently published study on the cryo-electron microscopy structure of the filamentous form of LACTB. From this and other studies, several specific properties of LACTB emerge, suggesting that the protein has distinct functions in different physiological settings. Resolving these issues by further research may ultimately lead to a unified model of LACTB’s function in cell and organismal physiology. LACTB is the only member of its protein family in higher animals and LACTB may, therefore, be of particular interest for future drug targeting initiatives.

## 1. Introduction

LACTB is a soluble protein localised in the mitochondrial intermembrane space (IMS) between the outer and inner mitochondrial membranes. LACTB can polymerise into stable straight helical filaments with a length of several hundred nanometres [1]. Recently, the cryo-electron structure of filamentous LACTB was resolved at a resolution of 2.8 Å [2], yielding unanticipated and important information about the catalytic mechanism and polymer formation. LACTB shares significant sequence similarity with bacterial penicillin-binding proteins and beta-lactamases (PBP-βLs) involved in peptidoglycan synthesis [3]. The proteins in this family harbour a strictly conserved serine residue essential for their catalytic activity. This serine residue is also conserved in LACTB, implying that LACTB is a catalytically active enzyme, although the physiological substrate(s) remain unidentified. Northern blot analysis indicates that LACTB is expressed in a majority of mammalian tissues [4].

Evidence indicating that LACTB is involved in lipid metabolism has accrued from several independent studies employing different experimental and bioinformatic strategies. Thus, an integrative analysis of gene networks for multigenetic traits detected a causal link between LACTB and increased omental fat mass [5,6]. This finding was validated in a transgenic mouse model showing that LACTB overexpression resulted in mild obesity [5,7]. In line with these findings, a GWAS study with global non-directed metabolomics demonstrated an association between LACTB and the metabolism of short fatty acids [8]. Moreover, two independent GWAS studies, aimed at identifying novel loci for blood lipid-related traits, revealed a link between LACTB and high-density lipoprotein metabolism [9,10]. LACTB has also been shown to be involved in the repression of the sterol regulatory-element binding protein (SREBP), which is a key transcriptional activator controlling the expression of genes involved in fatty acid and cholesterol biosynthesis [11].

A different manifestation of LACTB’s function was uncovered by the discovery that LACTB acts as a tumour suppressor in breast cancer cells [12]. In this study, a hypothesis was presented according to which the tumour suppressive effect of an increased LACTB level is mediated by a decreased activity of the mitochondrial enzyme phosphatidylserine decarboxylase (PISD). This enzyme catalyses the synthesis of the membrane phospholipid phosphatidylethanolamine (PE) from phosphatidylserine (SE). Since PE constitutes about 15–25% of the total cellular phospholipid pool, curbing the rate of PE formation from PS is anticipated to retard cell growth and proliferation, particularly affecting rapidly dividing cells such as cancer cells. 

Many challenging questions arise both regarding the biochemical function of LACTB and the potential role of LACTB dysfunction as a driver of pathological processes. Elucidating the biochemical and cell biological function of LACTB may hence lead to useful new insights into the regulation of the lipid metabolism and may enlighten on how aberrations in lipid metabolism contribute to the development of cancer, and possibly other disorders.

## 2. LACTB Is Localised to the Mitochondrial IMS

LACTB is encoded by the nuclear genome and imported to mitochondria, directed by an N-terminal 62 amino acid pre-sequence that is removed upon the import of the protein to mitochondria [1]. The location of the LACTB filaments between the outer and inner mitochondrial membrane is evident from electron microscopy images [1]. Earlier electron microscopy studies of mitochondria in situ in various organs have shown IMS filaments having the same size and shape as LACTB filaments, although the molecular nature of these IMS filaments was unknown at the time [13,14]. The 62 amino acid mitochondrial import sequence of LACTB (LACTB^1−62^) has been used to target several different fusion protein probes to the mitochondrial IMS. Using an engineered ascorbate peroxidase fused to LACTB^1−62^, in order to biotinylate the neighbouring proteins combined with analysis by mass spectrometry, resulted in the identification of 127 specific mitochondrial IMS proteins, thereby defining the IMS proteome [15]. Using LACTB^1−62^ fused to the photoactivatable fluorescent protein Dronpa, the localisation of mitochondrial nucleoids with respect to the IMS was determined by super-resolution microscopy [16]. Furthermore, LACTB^1−62^ fused to the photoactivatable fluorescent protein Eos was used to monitor changes in the IMS volume during hypoxic adaptation [17]. Finally, using LACTB^1−62^ fused to the bilirubin-binding fluorescent protein UnaB in combination with other site-specific probes, a mapping of the intracellular localisation of bilirubin was performed [18]. These studies indicate that LACTB is localised exclusively to the mitochondrial IMS, implying that earlier findings suggesting an association of LACTB with the mitochondrial ribosome, localised in the mitochondrial matrix space [19], can most likely be ruled out as an experimental artefact, possibly caused by the sedimentation velocity of polymeric LACTB filaments being similar to that of the subunits of the mitochondrial ribosome.

## 3. Structural Elements of LACTB

At least one homologue of LACTB is found in all branches of the animal kingdom, suggesting that LACTB is an indispensable protein for mitochondrial function in animal cells [3]. In many metazoans, the *LACTB* gene has undergone duplications, exemplified by flatworms which have a set of seven *LACTB* homologues [3]. The bacterial PBP-βLs from which LACTB has evolved are involved in the synthesis and metabolism of peptidoglycan, the principal component of the bacterial cell wall. Since eukaryotic cells lack peptidoglycan, LACTB must have been converted to accomplish some other biochemical function(s) during the early eukaryote evolution. Sequence alignments of LACTB with its bacterial homologues reveal that LACTB harbours not only the catalytic serine residue but also two other signature motifs, a [SY]X[NT]-motif and a -[KH][ST]G-motif, that together contribute to form the catalytic site [3]. In addition to the part of LACTB homologous to the PBP-βLs, LACTB contains two additional distinct structural features lacking from the bacterial homologues (Figure 1). Firstly, in LACTB, the PBP-βLs homology domain is interrupted in the middle by a 70-amino-acidslong segment, enriched in amino acids carrying charged side chains. The function of this middle region will be discussed below in the context of the cryo-electron microscopy structure of LACTB. Secondly, LACTB contains an N-terminal extension, which upon mitochondrial import is cleaved after G62 residue [1]. This cleavage leads to the display of an N-terminal tetrapeptide motif shared by two other mitochondrial IMS proteins, SMAC/DIABLO and Omi/HtrA2 (Figure 1) [20,21]. During apoptotic signalling through the intrinsic pathway, these two proteins are released from the mitochondria to the cytosol where they bind to the inhibitor-of-apoptosis (IAP) proteins [22] through the N-terminal tetrapeptide motif. The IAP proteins, such as X-linked inhibitory of apoptosis protein (XIAP), contain baculovirus IAP repeat domains which inhibit cellular caspases, including caspase-3 and -7 [23]. The interaction of SMAC/DIABLO or Omi/HtrA2 with XIAP can then lead to the activation of caspases, triggering the execution phase of apoptosis. The fact that a similar N-terminal tetrapeptide motif occurs on LACTB may imply that LACTB is involved in the modulation of apoptotic signalling events, as discussed further below.

## 4. LACTB and Lipid Metabolism—Anabolic or Catabolic Effect?

The first evidence that LACTB is connected to lipid metabolism came from analyses of quantitative trait loci and associated gene networks that, when perturbed by environmental factors, may lead to variations in disease traits [5,6]. In these untargeted studies, LACTB was identified together with ZFP90 and Lpl as a causative factor for obesity-related traits. This finding was then tested in a transgenic mouse model which, in agreement with the prediction, showed a steady gain in the fat-mass-to-lean mass ratio over wild type resulting in a 20% increase after a 14-week observation period [6]. This finding demonstrates that global LACTB overexpression leads to an overall anabolic effect on the lipid metabolism, resulting in an excessive accumulation of triglycerides. Independent evidence suggesting that LACTB acts in the setting of anabolic pathways came from the finding that LACTB expression is increased by insulin in human skeletal muscle, using a three hour hyper-insulinemic clamp to identify genes responding acutely to insulin signalling [24]. A somewhat different conclusion was drawn from studies of phospholipid metabolism in breast cancer cells, where it was demonstrated that increased LACTB expression leads to a diminished synthesis of the membrane lipid PE from PS through deactivation of the enzyme PISD [12]. A decreased availability of PE for membrane biosynthesis may become a limiting factor for cell proliferation in rapidly dividing cells, and such a mechanism would present an elegant explanatory model for the tumour suppressive effect of LACTB. Thus, in the context of tumorigenesis, LACTB exerted a growth-inhibiting and anti-anabolic effect, contrasting to the results obtained from mouse studies where an overall anabolic effect was observed. However, the reduction in PE synthesis induced by LACTB expression was observed exclusively in breast cancer cell lines and not in the corresponding non-tumorigenic mammary cell lines [12]. The cellular ATP level, ROS production and mitochondrial membrane potential remained unchanged upon LACTB expression. Therefore, the anti-anabolic effect of LACTB seen in tumorigenic cells might be a consequence of a reprogramming of the metabolic network specific to some tumorigenic cells, and not a manifestation of the normal function of LACTB in non-tumorigenic cells. Recent results from studies on LACTB expression in different cancer forms have yielded some support for this interpretation.

## 5. LACTB in Cancer—Context-Dependent Function?

Following the discovery that LACTB exerts a tumour suppressive effect in breast cancer cells, the expression level of LACTB and its effect on the cancer cell phenotype have been reported for several other cancer forms. While a majority of these studies support the concept that increased LACTB expression is connected with tumour suppression, some studies indicate that increased LACTB expression may instead exert a tumour promoting effect. In line with the original findings in breast cancer cells [12], a study employing breast cancer tissue microarrays demonstrated that LACTB downregulation correlated with poor clinical outcome [25]. Similar results were obtained for gliomas where the degree of LACTB downregulation correlated with an unfavourable overall clinical outcome, and in glioma cell lines LACTB overexpression resulted in an inhibition of cell proliferation [26]. In colorectal cancer, low LACTB expression level correlated with a poor clinical outcome, and an overexpression of LACTB in colon cancer cells suppressed proliferation, invasion and epithelial-to-mesenchymal transition [27,28,29]. Likewise, a low expression level of LACTB correlated with a poor prognosis in hepatocellular carcinoma and overexpression of LACTB in hepatocellular carcinoma cells resulted in the inhibition of proliferation, migration and invasion [30]. Similar results supporting a tumour suppressive effect of increased LACTB expression have been obtained for melanoma [31,32], gastric cancer [33,34], and lung cancer [35]. 

In contrast to the results of the above-mentioned studies, it was discovered that both the LACTB mRNA and protein expression levels were significantly increased in pancreatic cancer, and that a high expression level correlated with an unfavourable clinical outcome [36]. Similarly, it was found that the expression level of LACTB in nasopharyngeal carcinoma was elevated compared to the surrounding tissue, and that overexpression of LACTB in nasopharyngeal carcinoma cells promoted motility and increased metastasis capability, while knocking down LACTB in these cells had the opposite effect on the phenotype [37]. 

It is evident that the results of these studies cannot be reconciled with a clear-cut relationship between LACTB expression level and tumour aggressiveness, suggesting that the effect of increased or decreased LACTB expression on tumour cell phenotype depends on the underlying rearrangements in the lipid metabolism, and possibly other factors specific to each tumour type. Rapidly dividing cells have an increased demand for cellular building blocks, including fatty acids and cholesterol, and therefore the metabolic reprogramming occurring during any tumorigenic process must involve an enhanced lipid biosynthesis [38]. An upregulation of key enzymes in lipid biosynthesis and their transcriptional activators are, hence, of paramount importance in the tumorigenic process. In many cancer forms, an increased expression of ATP-citrate lyase, which directs substrate from the tricarboxylate cycle towards fatty acid biosynthesis and of the fatty acid synthase complex which condensates acetyl-coenzyme A units into palmitate, are considered metabolic hallmarks of the tumorigenic process [39]. An increased expression of hydroxymethylglutaryl coenzyme A reductase, the rate-limiting enzyme in cholesterol and isoprenoid synthesis, promotes cellular transformation and correlates with an unfavourable clinical outcome in breast cancer [40]. Moreover, a high level of the transcriptional activator SREBP is required to maintain fatty acid and cholesterol synthesis in glioblastoma, and a high expression of SREBP is associated with an unfavourable clinical end result [41]. Therefore, it is likely that the outcome of an increased or decreased LACTB expression in terms of tumour cell phenotype depends on a complex interplay between dysregulated metabolic enzymes and small molecule effectors, unique to each tumour type and stage of the tumorigenic process. As mentioned, and detailed further below, it is also possible that LACTB has biochemical functions besides influencing lipid metabolism, adding an additional level of complexity to the picture. Finally, it is worth noting that the observed changes in the LACTB expression levels in the clinical material could be either causative or reactive to the tumorigenic process.

## 6. Cryo-Electron Microscopy Structure of LACTB Filaments

The structure of polymeric LACTB was recently resolved at 2.8 Å resolution by cryo-electron microscopy [2]. Given the high sequence similarity of LACTB with class B low molecular PBP-βLs, it was not an unexpected finding that these proteins share a similar overall beta-lactamase fold, as previously indicated by homology modelling [1]. The N-terminal 40 amino acid segment and the 70 amino acid middle region of LACTB remained incompletely resolved in this cryo-electron microscopy analysis. The organisation of the conserved catalytic amino acids motifs into the active site follows the same principle as that of the PBP-βLs (Figure 2). Importantly, it was found that the filament scaffold formation of LACTB takes place through interactions between amino acids in the beta-lactamase domain, and not through the middle region as proposed earlier [1]. The overall structure of the LACTB filament consists of two antiparallel chains forming a right-handed helix where interaction interfaces are formed between successive monomers on the same strand, orthogonal monomers on separate strands, and diagonal monomers on separate strands. Multiple-site mutations on these interfaces on the beta-lactamase domain are required to disrupt filament formation. In this structure, the middle region is facing outwards from the longitudinal axis, possibly contributing to allow for conformational flexibility during the catalytic process [2]. Deletion of the middle region has little impact on the structure of the beta-lactamase domain and does not significantly impede filament formation.

The structure also shows that the catalytic centre formed by the three signature motifs is partially covered by a positive loop, also interacting with the middle region protruding from a nearby monomer. Two putative substrate and product delivery tunnels, formed between successive dimers in the helix were identified, having dimensions allowing for the entry and exit of at least short peptides. Interestingly, deletion of the middle region has little effect on helix formation and structure, but results in a complete abolishment of the catalytic activity using the test substrate Ac-YVAD-AMC. As predicted, single-site amino acid mutations in the loop region result in diminished test substrate cleavage activity. Multiple-site mutations of residues in the beta-lactamase backbone, forming interaction interfaces between monomers, result not only in an impaired filament formation but also largely reduce the catalytic activity, implying that polymerisation is necessary for the catalytic activity. Bacterial PBP-βLs are not known to polymerise into high-order filaments, and therefore the capability of filament formation must have been acquired during the evolution of LACTB. This implies that the enzymatic activity of the monomeric protein was gradually switched off during evolution in order to require reactivation through polymerisation and filament formation. Further insight into these intriguing matters will require an identification of the physiological substrate(s) of LACTB. Nevertheless, the results of this important study indicate that LACTB monomers are able to auto-polymerise to form stable rigid helical filaments, and provide a structural basis for filament-formation demonstrating that polymerisation is mediated by the beta-lactamase backbone.

## 7. Perspectives and Future Directions

In this review we have summarised the current knowledge about LACTB, and based on this we will highlight some prominent matters that, in our opinion, merit particular attention and may generate significant new insight about the function of LACTB. Current understanding of LACTB allows us to define three functional aspects of the protein: (a) the active-site serine protease/hydrolase; (b) the property of homopolymer formation; and (c) potential binding to IAP proteins through the N-terminal tetrapeptide motif. Further investigations into these three properties can help to clarify the functions of LACTB in cells under different physiological and pathophysiological conditions. 

To resolve the question of the enzymatic substrate(s) of LACTB, it may be useful to consider the unique evolutionary history of the protein, being the only member of the PBP-βLs protein family in higher eukaryotes. During completion of the endosymbiotic process, the resulting primitive eukaryotic cells dumped the whole enzymatic machinery for peptidoglycan synthesis, including PBP-βLs proteins, while retaining the ancestral *LACTB* gene in the genome. This scenario suggests that the ancestral LACTB protein had already been adopted for a function distinct from peptidoglycan synthesis in the early eukaryote. Some inklings about the nature of the physiological substrate of LACTB are provided by the structure of LACTB’s substrate binding pocket, and by the finding that catalytic activity depends on polymerisation into filaments. Given the mentioned evolutionary and structural constraints, it is not likely that the physiological substrate of LACTB is another protein molecule, but is more likely to be a peptide or smaller metabolite. Some proteins of the PBP-βLs family show hydrolytic activity towards esters rather than peptide bonds [44], implying that the possibility of a lipid being the physiological substrate of LACTB cannot be excluded. However, a direct enzymatic role of LACTB in lipid turnover in some major pathway of lipid metabolism seems unlikely, at least in non-tumorigenic cells [12]. 

The cryo-electron microscopy analysis indicated that polymerisation of LACTB is necessary for its enzymatic activity and that polymerisation is mediated by amino acid residues localised in the beta-lactamase domain [2]. In particular, substitution mutations of amino acid residues E149, R151, D355, and Y473 obstructed the assembly of LACTB monomers into filaments without affecting the beta-lactamase structure itself [2]. Inspection of the sequence alignments of LACTB orthologs in eukaryotes reveals that these amino acid residues are conserved from *Dictyostelium* to human [3], suggesting that polymerisation is a primitive property linked to the regulation of the enzymatic activity. LACTB filaments in situ in the IMS appear to be tethered to the mitochondrial membrane at both ends [1], prompting the question as to whether the ends of the LACTB filaments bind directly to membrane phospholipids or to some, as yet unidentified, membrane-anchoring protein(s). A related question meriting attention is whether LACTB can depolymerise in situ and whether polymerisation–depolymerisation cycles form part of the physiological function of LACTB, as is the case for several other proteins forming homopolymers, such as actin and tubulin.

Whether LACTB is released from mitochondria in parallel with cytochrome c, SMAC/DIABLO, and Omi/HtrA2, to modulate caspase activity though IAP-binding during pro-apoptotic signalling is a question deserving urgent attention. In this simplified scenario, disregarding any effects of LACTB on lipid metabolism, a higher expression level of LACTB would be anticipated to promote caspase activation and hence to have a pro-apoptotic effect. Such a mechanism could offer an additional explanation for the tumour suppressive effect of LACTB, observed in several cancer forms. Whether such interaction between LACTB and IAP-proteins requires the depolymerisation and release of the whole LACTB protein from the IMS to the cytosol is a key question. It is worth noting that an interaction between the N-terminal segment of LACTB with IAP-proteins does not necessarily require release of the whole LACTB protein from mitochondria, but could well be accomplished though cleavage of a portion of the N-terminus which would then diffuse across the outer mitochondrial membrane through the voltage-dependent anion channel (VDAC) to bind to IAP-proteins in the cytosol. IAP-proteins are frequently overexpressed in cancers leading to suppression of pro-apoptotic signals and resulting in an increased tumour aggressiveness [45]. Several small molecules mimicking the IAP-binding tetrapeptide motif of SMAC have been designed and are currently in clinical trials for both solid tumours and blood malignancies [46]. A deeper understanding of IAP-mediated mechanisms leading to an increased tumour aggressiveness, including the potential role of LACTB therein, may contribute to leverage therapeutic strategies involving IAP-binding molecules.

An identification of the physiological substrate(s) of LACTB and a clarification of the regulation of its polymerisation and possible depolymerisation promises to yield significant novel insight into the physiological function of LACTB. Determining whether the putative IAP-binding tetrapeptide motif of LACTB fulfils a similar function in apoptotic signalling as the tetrapeptide motif of SMAC/DIABLO and Omi/HtrA2, or whether this motif has a function outside the apoptotic signalling cascade will yield essential novel information not only about the function of LACTB but may also be useful for the design of future IAP-antagonists.

In conclusion from our analysis, it follows that LACTB may exert distinct functions in different physiological settings, a primitive function pertinent to lipid metabolism conserved from the PBP-βLs protein, and an acquired function in apoptotic signalling adopted during the evolution of higher animals. Ancient proteins having acquired an additional function during the course of evolution are not without precedent, as exemplified by tryptophanyl-tRNA synthetase. Moreover, a dual role of LACTB in lipid metabolism and apoptotic signalling could provide a framework to interpret and explain the contradictory outcomes of LACTB expression in different cancer forms. The recent elucidation of the structure of LACTB together with emerging new concepts and data can give valuable insight into the biochemical and physiological functions of LACTB in cells.

## Figures and Tables

**Figure 1 cells-11-01703-f001:**
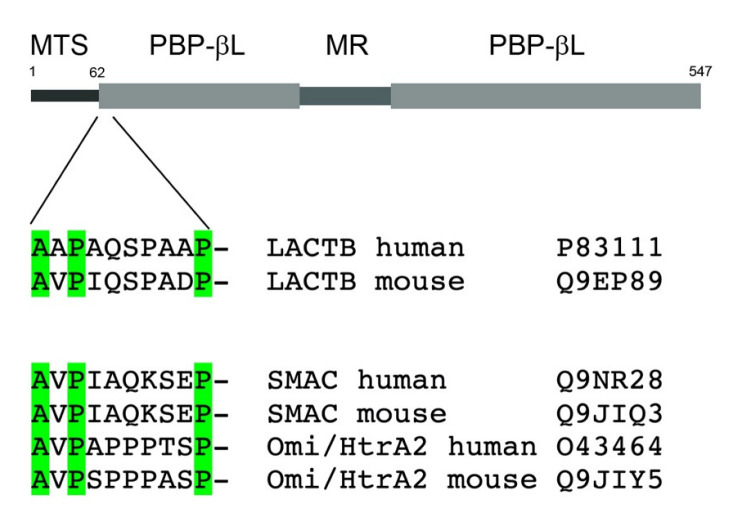
Structural elements of LACTB: MTS, mitochondrial targeting signal; PBP-βL, penicillin-binding and beta-lactamase homology domain; MR, middle region. The location of the potential IAP-protein binding N-terminal peptide motif in LACTB is indicated. Below a sequence comparison of IAP-protein binding motifs in the mitochondrial intermembrane space proteins SMAC and HtrA2.

**Figure 2 cells-11-01703-f002:**
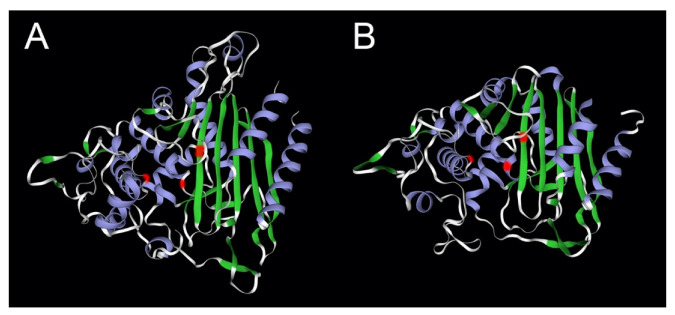
Comparison of the beta-lactamase fold of LACTB with that of the bacterial PBP-βLs protein flp. Panel (**A**): LACTB (P83111); panel (**B**): flp protein of *Staphylococcus aureus* (Q7A5Q5). The locations of the catalytic site residues in the signature motifs [3] are marked in red, -**S**ISK-, -**Y**ST-, and -**H**TG- for LACTB, and -**S**NTK-, -**Y**SN-, and -**H**SG- for the fpl protein. The structure of LACTB was determined by cryo-electron microscopy [2] and that of flp by X-ray crystallography [42]. The image was generated partially using SWISS-MODEL [43].

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
