# Peer review of "Unveiling the Function of the Mitochondrial Filament-Forming Protein LACTB in Lipid Metabolism and Cancer"

_cells, 2022, doi:10.3390/cells11101703_

Round 1

Reviewer 1 Report

In the current manuscript, Cascone et al extensively analyze knowledge on the mitochondrial protein LACTB. LACTB is a filament-forming molecule involved in lipid metabolism and in tumorigenesis. In spite of recent insights in its structure and function, which suggest that LACTB could play important functions, research  remains insufficient to draw a comprehensive picture of LACTB biology. By assembling a well-organized manuscript, the Authors highlight the need of further investigations aimed at dissecting LACTB interactions, regulators and biochemical functions, connecting them with the structural features of the protein and with specific pathophysiological conditions, such as neoplastic progression.

I have a couple of minor points:

  • line 47: it is difficult to envision that LACTB acts as a SREBP repressor, as this suggests a direct interaction. I suggest to use an indirect statemetn such as "it is involved in the repression"
  • - line 176: any effect of LACTB on tumor cells not necessarily depends on its role(s) in regulating lipid metabolism. Even if this is suggested at line 196, it should be more explicitly stated here.
  • - line 300 and following: a role of LACTB on IAP regulation is only proposed. So, speculation on the importance of IAP-directed therapies is a bit out of focus.

Author Response

Response to reviewer 1.

We thank reviewer 1 for the valuable comments on the manuscript. We have addressed the issues raised as follows:

Minor points.

Line 47. Response: changed according to suggestions by reviewer.

Line 176. Response: text has been added to clarify that any effect of LACTB does not necessarily depend on changes in the lipid metabolism.

Line 300 and following: the text has been condensed and one reference has been deleted.

Reviewer 2 Report

This review summarizes the current knowledge about LACTB, a mitochondrial intermembrane space protein that is evolutionarily related to the bacterial penicillin-binding/β-lactamase protein family involved in peptidoglycan synthesis. Studies have suggested its role in lipid metabolism, and tumor suppression function of increased LACTB expression have also been reported for several cancer cell lines. Furthermore, the authors proposed that it might be involved in the modulation of apoptotic signalling events as LACTB shares a similar N-terminal tetrapeptide motif with SMAC/DIABLO, a proapoptotic protein that increases caspase activation by binding to the inhibitor of apoptosis proteins (IAPs). However, its precise biochemical function remains to be elucidated.

Overall, I find this manuscript very well written and well-organized. It is a pleasure for me to read it. I just have 2 small suggestions:

  • Could the authors mention it somewhere in the text about the expression patter of LACTB? Is it ubiquitously expressed? Are there reports on expression levels in different tissues or under stress conditions?
  • The authors focus on studies with positive findings. I think negative results, such as there was no report of an effect of LACTB overexpression in normal cells, and no significant changes in ATP, ROS, membrane potential, and mitochondrial structure were observed after induction of LACTB expression, might also be worth mentioning.

Author Response

Response to reviewer 2.

We thank reviewer 2 for the valuable comments on the manuscript. We have addressed the issues raised as follows:

Minor points.

Expression pattern of LACTB.

To the best of our knowledge, the expression of the LACTB protein in various organs and tissues has not been systematically quantified thus far. However, northern blot analyses of mouse and human tissues indicate that LACTB mRNA is expressed in a majority of the tissues. Text has been added on lines 37-38, including one new reference. As detailed in section 5, LACTB is also present in a number of tumour types and their corresponding normal tissues. Changes in LACTB expression under various stress conditions, have to our knowledge not been systematically studied.

Concerning negative findings on ATP, ROS, membrane potential, and mitochondrial structure in cells after induction of LACTB. Response: text has been added on lines 159-160.